# Novel Processing Algorithm to Improve Detectability of Disbonds in Adhesive Dissimilar Material Joints [note 1]

**DOI:** 10.3390/s21093048

**Published:** 2021-04-27

**Authors:** Damira Smagulova, Liudas Mazeika, Elena Jasiuniene

**Affiliations:** 1Ultrasound Research Institute, Kaunas University of Technology, K. Barsausko Str. 59, LT-51423 Kaunas, Lithuania; liudas.mazeika@ktu.lt (L.M.); elena.jasiuniene@ktu.lt (E.J.); 2Department of Electronics Engineering, Kaunas University of Technology, Studentu St. 50, LT-51368 Kaunas, Lithuania

**Keywords:** NDT, ultrasonics, adhesive bonding, dissimilar materials, sensitivity analysis, detectability

## Abstract

Adhesively bonded dissimilar materials have attracted high interest in the aerospace and automotive industries due to their ability to provide superior structural characteristics and reduce the weight for energy savings. This work focuses on the improvement of disbond-type defect detectability using the immersion pulse-echo ultrasonic technique and an advanced post-processing algorithm. Despite the extensive work done for investigation, it is still challenging to locate such defects in dissimilar material joints due to the large differences in the properties of metals and composites as well as the multi-layered structure of the component. The objective of this work is to improve the detectability of defects in adhesively bonded aluminum and carbon fiber-reinforced plastic (CFRP) by the development of an advanced post-processing algorithm. It was determined that an analysis of multiple reflections has a high potential to improve detectability according to results received by inspection simulations and the evaluation of boundary characteristics. The impact of a highly influential parameter such as the sample curvature can be eliminated by the alignment of arrival time of signals reflected from the sample. The processing algorithm for the improvement of disbond detectability was developed based on time alignment followed by selection of the time intervals with a significant amplitude change of the signals reflected from defective and defect-free areas and shows significant improvement of disbond detectability.

## 1. Introduction

Adhesive bonding, mechanical fastening, and welding are the joining methods that have reached popularity in aerospace and automotive industries. A major problem of mechanical joining such as fastening is the concentration of high level of stress around the holes of fasteners. This limitation leads to more complex and severe deterioration in the strength of bonding of composite structures compared to metal ones [1,2,3,4]. In the case of adhesive bonding, the high stress concentration is eliminated, and uniform stress distribution is provided. Other advantages of adhesive bonding are its fatigue resistance, as well as the ability to preserve the structural integrity of joining materials, join dissimilar materials, and reduce the weight of the structure. Due to growing concern in fuel consumption and pollutant emissions, adhesive bonding technology is widely used more and more in aerospace and automotive industries [5,6,7]. Furthermore, metal components are being substituted by composite materials, specifically by carbon fiber-reinforced plastic (CFRP), to lighten the load-bearing structures and save energy. CFRPs are being integrated due to their high strength to weight ratio, high stiffness to weight ratio, low density, and high mechanical properties. Consequently, a better performance of CFRP structures is provided in constructions [7,8]. However, not all metal components can be replaced by CFRPs or other light composite materials, because of the specific requirements of the construction, low bearing strength and stiffness, dependence on laminas configuration, and environmental conditions, which characterize the mechanical properties of the CFRPs. In that case, either dissimilar light materials such as different aluminum alloys joined using friction stir welding with nanoparticles fillers to enhance the properties of the joint are used [9] or advanced joints of dissimilar materials metal bonded to composite are in use and lead to the structure improvement. CFRPs and aluminum alloys are more attractive lightweight materials, balancing costs and performance [5,8]. 

Composite and metal components with desired material characteristics can be joined in order to generate unique characteristics such as improved strength and stiffness as well as damage resistance from dissimilar material joints after combination. CFRP/aluminum and CFRP/titanium joints are used in airplane engine cowlings, wing panels, fixed trailing edges, fairings, and other parts of the construction [10]. Due to the use of dissimilar joints in such expensive constructions, the regular inspection is required to detect defects and damages, which can be critical to the structure as well as the assessment of structure suitability for further safe use. Costs, people, life, and health safety are dependent on the structural health of these constructions [8]. 

Adhesive degradation can significantly reduce the bonding strength of the structure of adhesively bonded dissimilar joints. Environmental conditions and stresses influence the integrity of adhesive bondline of joints and lead to the appearance of various types of defects [6,11,12,13]. The main problem of such structures is debonding between adhesive and adherends. The detection of adhesive failure in bonded similar materials is a challenging process because of complex interfacial location and thin adhesive layer thickness [8,14]. There are some works that propose traditional ultrasonic, advanced nonlinear ultrasonic, laser ultrasound, acoustic, guided waves, eddy current, thermography, shearography, X-ray tomography, and other non-destructive methods to evaluate the bondline integrity of the structures [6,15,16,17,18,19,20,21,22,23]. Yilmaz et al. [14] was using advanced ultrasonic non-destructive testing for the detection of weak bonds in composite–adhesive structures. High frequency and high-resolution acoustic microscope were used for the inspection of quality of adhesive layer. Additionally, for better quality visualization, a shape-based feature extraction post-processing algorithm was developed. In other work, Yilmaz et al. [24] proposed the fusion technique of ultrasonic testing and thermography data to evaluate bonding quality.

The process of detecting such defects becomes even more complex in joints of dissimilar materials. Due to the fact that composites and metals have quite different acoustic impedance, it complicates the determination of the presence of a defect between two dissimilar materials [4,8]. There is a lack of works carried out on the study of integrity of adhesive bonding between dissimilar material joints. From recent works, Sun et al. [25] proposed electromagnetic-pulse-induced acoustic testing for the detection of disbonds, which generates guided waves for inspection of hydrogen tanks made of bonded composite to metal. This method uses pulsed current and pulsed magnetic fields to be able to excite guided waves. The intensity can pass through a very thick layer of composite material and induces guided waves in the inner metal layer. Jahanbin et al. [26] applied the ultrasonic guided waves method for testing the bondline quality of hybrid joints. However, the low-frequency range used for the excitation of guided waves to travel a long distance and avoid the loss of energy is limiting the method application in short area components. In addition, the formation of interface guided waves in joints is complicated. There is a possibility of interference of waves generated by reflections from component boundaries and interface waves. The defect position at or near the interface is limiting the use of the technique as well. Moradi et al. [27] applied thermography for edge disbond detection of carbon/epoxy in a repaired aluminum structure. Since inspection using flash thermography is a challenging issue, several image processing methods were presented such as Fourier transform and Daubechie’s wavelet transforms. As a result, the enhanced infrared images were received. Jasiuniene et al. [28] proposed a novel signal post-processing algorithm to reconstruct bonding area after ultrasonic inspection of complex titanium bonded to carbon fiber composite component. The proposed algorithm provided the ability to detect defects as well as estimate the position and depth. There are some works where ultrasonic, thermography techniques as well as their fusion were applied [29,30,31]. Nevertheless, the question of detecting defects in dissimilar material joints remains relevant and requires more study and research. 

The aim of this work is to improve disbond detectability and location in adhesively bonded dissimilar joint between aluminum and CFRP. The task is complicated by the fact that common disbonds are located inside the adhesive layer between two adhesive tapes. Therefore, the defects are similar to cohesive failure. The component of dissimilar material joints with three artificial defects in the bondline were investigated using the immersion pulse echo technique and focused transducers. The tasks were to perform simulations of the inspection to study the behavior of ultrasonic waves in the joints, perform quantitative evaluation in order to determine the most influential parameters on the inspection detectability, perform experimental inspection and modeling of the signals based on material properties for further comparison of the results, and study the boundary characteristics and development of post-processing algorithms for defect detectability improvement. 

## 2. Materials and Methods

In this section, the characteristics of dissimilar material joints and investigation methodology are described. Inspection simulation, sensitivity analysis, evaluation of boundary characteristics, and modeling of the signals reflected from the sample boundaries were performed in order to study the complexity of disbond detection in the adhesive layer, identify influencing factors, as well as develop an investigation methodology and post-processing algorithm to improve detectability.

### 2.1. Component Description

A single-lap joint of adhesively bonded CFRP and aluminum is under investigation. The scheme and photo of the joint of dissimilar materials is shown in Figure 1. 

The thickness of aluminum is 1.61 mm and the thickness of CFRP is 5.11 mm. The CFRP plate was made of 41 unidirectional prepregs; the first and the last layers are glass prepregs. We used 3 M Scotch-Weld AF163 k-red adhesive tape to glue dissimilar plates. There are 3 artificial disbonds between 2 layers of adhesive tapes made of 2 layers of release film. The thickness of the adhesive layer is 0.22 mm. Therefore, theoretically, it was assumed that the thickness of single adhesive tape is 0.11 mm. Disbonds of three different dimensions are located at one side of the component while the other side represents perfect bonding. 

### 2.2. Modeling of Sensitivity Analysis and Design of NDT Technique

#### 2.2.1. Inspection Simulation 

Inspection simulation of the sample was performed to study the behavior of ultrasonic waves and their propagation through the layers of adhesively bonded dissimilar materials. In addition, A and B-scans of the simulations were visualized to evaluate and compare amplitudes of the reflections. The results of inspection simulations were used for algorithm development to improve the detectability of the NDT technique. The whole inspection process was simulated in CIVA software developed by CEA (French Atomic Energy Commission). 

Three options of disbond location (between aluminum and adhesive, in the middle of adhesive, and between adhesive and CFRP) were modeled to compare and study the detectability of defects [8]. The smallest defect with the size of 5 × 5 mm was selected for the investigation. Immersion pulse echo inspection using a 10 MHz focused transducer was simulated. The focal distance in water is 50.8 mm, and the diameter of the transducer is 9.75 mm. The higher frequency transducers were not selected in order to avoid attenuation of the signal in the adhesive layer and to be able to detect disbonds in the epoxy. This method is designed to avoid very expensive inspections, such as acoustic microscopy, and make it conventional and more suitable for industrial use. The distance between the transducer and surface of the object was calculated according to Equation (1) so that the focus is on the interface of dissimilar material joints: (1)$$W_{p}=F-M_{D}\left(\frac{V_{tm}}{V_{w}}\right)\;,$$
where *F* is the focal distance, $M_{D}$ is the material depth, $V_{tm}$ is the velocity of ultrasound in the test material, and $V_{w}$ is the velocity in water.

As a result, the distance between the transducer and the component is 43.5 mm. The inspection simulation set-up is shown in Figure 2. Scanning was performed with a 1 mm step size. 

The resulting B-scans of three different options of defect location—between aluminum/adhesive, in the middle of adhesive, and the adhesive/composite—are shown in Figure 3. 

From the presented images, it follows that in the case of the presence of a disbond, multiple reflections can be observed. These multi-reflections enable distinguishing the position where the disbond is present. On the other hand, the depth in the adhesive layer has an impact on the detectability due to attenuation in the adhesive layer and the overlapping of signal reflections from the adhesive top, bottom, and disbond [8,17,32,33,34,35,36]. The effect of overlapping depends on the thickness of the adhesive layers, defect depth location, and selected frequency. However, it seems that the most promising disbond detection and assessment technique should be based on the analysis of these multiple reflections. As a result, the analysis of multi-reflections can be used in the development of a processing algorithm for detectability improvement. 

#### 2.2.2. Determination of the Influential Parameters 

Sensitivity analysis was performed in order to determine the most influential parameters on disbond detectability in a layered structure. These parameters are taken into account for further development of the technique to improve detectability. CIVA software provides the possibility of sensitivity analysis avoiding the experimental approach, which requires large-scale tests leading to high costs and time consumption [37,38]. 

Using numerical simulation, the number of influential parameters on which the detectability of the defect depends can be introduced for analysis [37,39]. 

A study of the sensitivity of varying parameters consists of two main parts such as calibration case and metamodel computation.

Calibration of inspection simulation was performed to determine the maximum amplitude of the signal reflected from the debonding type defect for further use in metamodel calculations as a reference amplitude. The inspection of the specimen of adhesively bonded dissimilar materials using a 10 MHz focused transducer was modeled. A rectangular-type defect placed between the aluminum and adhesive layers was selected to simulate disbond and to perform calibration of inspection simulation. According to the structure of the object and the selected non-destructive technique, the detection threshold in this case is dependent on the contrast of amplitudes between the defective area echo and the not defective (“Healthy”) area echo [38]. The set-up and B-scan of inspection computation is shown in Figure 4. 

The reference value is the maximum amplitude of the signals reflected from the defect that was used in metamodel analysis.

The metamodel approach provides the quantification of parameters influencing the inspection results by establishing a number of test cases and interpolation functions to compute the metamodel [40,41]. Metamodel estimation gives access to a large number of results in the defined range of varying parameters. It is known from the component description subsection that during sample manufacturing, the disbonds made of release film were placed between two adhesive tapes. However, the manufacturing process of the sample has a number of influencing factors on the location of disbonds in the adhesive layer. Therefore, the actual position of the defects in adhesive can differ from that stated in the description. According to the inspection technique and object structure, the following influential parameters were identified: aluminum longitudinal wave velocity, defects dimension along the bondline *x* axis, defects depth in the adhesive layer, water path between the transducer and object surface, thickness of the aluminum layer, and the incidence angle (the angle of the transducer beam according to the surface of the sample). 

The incidence angle was taken into account in order to study the influence of the object curvature along *x* and *y* axes on the detectability. Variation ranges for each influencing parameter as well as statistical distribution laws were defined and presented in Table 1. Statistical analysis was performed to identify the maximum and minimum values of wave velocity, thickness of the aluminum layer, and the water path between the transducer and object surface. The incidence angle range was determined while adjusting the beam angle of transducer to the surface of the component in the experimental process. The variation range of defect depth in the adhesive layer was limited by the thickness of the layer and was selected appropriately. The lateral defect dimension was selected to change in the range between 0.25 and 5 mm. The dispersion of values was characterized by standard deviation without bias. The total number of 768 simulations was performed to build a consistent metamodel.

The Sobol indices statistical method was used to compare the impact of selected parameters on the output results. The values of sensitivity as well as their proportion of impact for each variable in percentage is presented in Table 2.

The Sobol indices diagram of sensitivity analysis clearly demonstrates that there are three critical parameters with a high impact (Figure 5): defect depth in the adhesive layer, and the incidence angle and thickness of the aluminum layer. The possible uncertainty in the knowledge of the exact ultrasound velocity in aluminum and the water path or the distance between the transducer and the component surface has quite a low influence on the detectability of the technique.

One of these influencing factors—depth in the adhesive layer—is the parameter of the defect and, of course, it cannot be optimized or adjusted during inspection. On the other hand, it demonstrates that there is a promising possibility for disbonds depth assessment. Two other parameters are related directly to the inspection set-up and indicate where the main attention should be paid in the experiment adjustment or even the development of an inspection technique in general. The incidence angle or curvature of the sample is one of the major influential parameters. Therefore, alignment of the signals reflected from the component surface according to the arrival time has a high importance and would enable eliminating the impact of curvature and increase disbond detectability. 

#### 2.2.3. Evaluation of Boundary Characteristics and Multiple Reflections

Boundary characteristics of the component were analyzed to understand the expected behavior of the propagating ultrasonic waves through the layers in the case of different disbond location in the adhesive layer and evaluate theoretically the amount of energy of multiple reflections, which will be received by the transducer. To achieve this goal, the acoustic impedance of each material as well as the reflection ($K_{R}$) and transmission ($K_{T}$) coefficients were calculated.
(2)$$Z_{k}=V_{k}\cdot \rho_{k},$$
where $V_{k}$—ultrasound velocity in materials of the sample, $\rho_{k}$—density of materials, and *k* denotes different types of material.
(3)$$K_{R}=\frac{Z_{2}-Z_{1}}{Z_{2}+Z_{1}},$$
(4)$$K_{T}=\frac{2\cdot Z_{2}}{Z_{2}+Z_{1}},$$
where $Z_{1}$ and $Z_{2}$ are acoustic impedances of the 1st and 2nd media along the wave propagation path. 

Since the interface of the component is under interest, three options were modeled: the interface without any defect (perfect bonding), defect location between aluminum and adhesive layers, and defect location in the middle of the adhesive layer. The amplitude of reflections from different boundaries considering multiple reflections are estimated according to Equation (5) for the model of aluminum/air boundary and Equation (6)—for the adhesive/air and adhesive/GFRP boundaries:(5)$$A_{1n}=K_{T12}\cdot K_{T21}\cdot K_{R23}^{n}\cdot K_{R21}^{\left(n-1\right)},$$
(6)$$A_{2n}=K_{T12}\cdot K_{T23}\cdot K_{T32}\cdot K_{T21}\cdot K_{R34}^{n}\cdot K_{R32}^{\left(n-1\right)},$$
where *n* is a number of multiple reflections from particular boundaries (degree), *K_T_* is a coefficient of transmission reflection, *K_R_*—reflection coefficient, and the numeric indices—component layer numbers.

Ultrasonic wave velocity in aluminum, CFRP, and adhesive layers of the component was measured using the ultrasonic pulse-echo method in different measurement systems as Omniscan, TecScan, and Acoustic Microscopy using 10, 15, and 50 MHz transducers. As a result, it was determined that ultrasound velocity in aluminum is 6363 m/s, in CFRP, it is 2800 m/s, and in adhesive, it is 2315 m/s. The propagation time in each of the layers can be estimated according to:(7)$$t_{k}=\frac{2\cdot H_{k}}{V_{k}}\;,$$
where $H_{k}$ is the thickness of material layers, $V_{k}$ is an ultrasonic wave velocity in the material of the layer, and *k* is a type of material.

The possible paths of wave propagation are shown in Figure 6. Air medium characterizes disbond. The time interval between multiple reflections from the aluminum/adhesive boundary is 0.5 us. The example of calculation of multi-reflection time propagation in a particular layer *t_nk_*:(8)$$t_{nk}=\overset{K}{\underset{k=1}{\sum }}\overset{N_{k}}{\underset{n=1}{\sum }}\frac{2\cdot H_{nk}}{V_{nk}},$$
where *n* = from 1 to *N* is the number of multiple reflections from a particular boundary, *k* is a particular layer number, $H_{nk}$ is the thickness of the *K*th layer, and $V_{nk}$ is an ultrasonic wave velocity in the *K*th layer.

Three modeled options of perfect bonding and total debonding in order to calculate the energy received by the transducer are shown in Figure 7. The propagation paths of ultrasonic waves through the layers of modeled objects as well as required for calculation reflection and transmission coefficients are illustrated in the set-up for each case. The air medium characterizes total debonding. Multiple reflections from aluminum/air, adhesive/air, and adhesive/GFRP were investigated. A theoretical pulse response in a layered structure of each reflection from the boundaries of the component is also shown in the figure. The plots with impulses are characterizing the arrival time of reflections from different structure layers, where H(t), a.u. is the arbitrary unit of the transfer function. For each case of boundary as aluminum/air, adhesive/air, and adhesive/GFRP multiple reflections, t_1i_–t_4i_, t_1ad,air_, and t_1ad,GFRP_ are plotted. The polarity of pulses is dependent on the acoustic impedances of materials of the layered structure and the change of phase. Due to these factors, polarity can be positive or negative.

Amplitude of multiple reflections from the object boundaries for the case of aluminum/air, adhesive/air, and adhesive/GFRP were calculated based on the boundary characteristics of the layered structure. The main idea of the analysis of multiple reflections is the first reflection from a particular boundary to be assessed is not the most sensitive but rather the repeated subsequent reflections. The bigger the number of reflections, the more information about boundary conditions is acquired [17]. It can be illustrated by the amplitudes of different reflections calculated according to Equations (5) and (6) presented in Table 3. 

For example, to compare amplitudes of the first reflection A_1_ on the boundaries aluminum/air and adhesive/air, the ratio is close to 2 (6 dB). However, in the case of fourth reflection A_4_, the same ratio is already 3.3 (10 dB). So, it can be also concluded that in order to assess bonding conditions, it is necessary to analyze a possibly bigger number of multi-reflections. The effectiveness of analysis of multiple reflections is quite high and can significantly improve detectability. 

Additionally, analyzing the disbond location in the sample, it can be observed that the detectability of disbonds located between aluminum and adhesive with A_1_ of −0.2915 is almost 2 times higher compared to the disbond location between two tapes of adhesive with A_1_ of −0.1451 [8]. In the case of defect location in the middle of the adhesive layer, attenuation is slightly higher. Values of multiple adhesive/air reflections are very low, which makes disbond detectability more complicated, taking into account all the influencing factors studied.

#### 2.2.4. Modeling of the Signal in a Layered Structure

The layered structures inspection technique based on the model analysis of multiple reflections from different interfaces of the sample of adhesively bonded dissimilar materials was performed using a custom developed program using MatLab software. The obtained results were analyzed in order to study signal propagation through the layers of the sample, signal form change due to the reflections from the boundaries of thin materials, and the influence of reverberations present in the signal received experimentally. Additionally, the aim of signal modeling is to create the signal close to the experimental one for further comparison of modeled signals reflected from defective and not defective areas in order to determine time instances with the most significant amplitude changes. The idea of selecting time intervals with the most significant amplitude change for visualization on C-scans can provide an improvement of detectability.

In order to carry out simulation as close as possible to the experiment, the reference signal of experimentally measured reflection from the flat aluminum block was used. 

The model parameters have been optimized manually in software in order to achieve the best fit between modeled and experimental signals. Optimization was performed by selecting the value of parameters such as material thickness, density, and ultrasound velocity, which provides the best match to the signal obtained experimentally. The range of parameters values does not exceed acceptable ones, both measured and from references. The final results of values after optimization are presented in Table 4. The time of flight of each reflection from the sample boundaries was calculated according to Equation (7). The acoustic impedance of each material was calculated according to Equation (2).

The signal reflected from the layered structure of the sample was modeled according to:(9)$$u\left(t\right)=y_{ref}\left(t\right)⊗h\left(t\right),$$
where $y_{ref}$ is a signal that was measured using a reference block, and h(t) is the theoretical pulse response in the layered structure.

As a result, the modeled and experimental signals reflected from the adhesively bonded dissimilar materials were compared and analyzed. In addition, the signal reflected from the defect (air) as shown in Figure 7b was modeled and compared to the signal of perfect bonding of the sample. The comparison of signals is required for the determination of differences between two signals and further development of the method for disbond detectability improvement.

### 2.3. Proposed Algorithm for the Improvement of Detectability

The inspection object usually is not ideally flat. Moreover, conventionally, some misalignment of scanning plane compared with the sample plane is present. This causes problems for the accurate determination of time gate windows corresponding to different reflections. As a result, the detection of disbond type defects becomes more complicated. The sample, which is under investigation, is not perfectly flat, but it is curved along *x* and *y* axes. In order to overcome this problem and improve the detectability of disbonds in dissimilar material joints, an advanced signal processing algorithm was developed. The functional block diagram of the algorithm is shown in Figure 8. The main steps of the algorithm are as follows:
Filtering of the signals;The alignment of the signals with respect to surface reflection;Setting the time gate at signal reflection from the surface of the component;Determination of time gate windows with high change in amplitudes for multiple reflections from the interface;Calculation of peak-to-peak amplitudes in selected time intervals;Calculation of ratio of peak-to-peak amplitudes in selected time intervals in order to evaluate the decay of multiple reflections.

In order to achieve as much as possible accurate alignment, at first, the signals have been filtered. For signal denoising, different filters can be used [24,42]; in this work, band-pass filter was used (Figure 9). Broadband filter characteristics were taken to avoid the loss of useful data of the signal reflected from thin layers of adhesively bonded dissimilar materials and enhancement of overlapping effect. The bandwidth of the bandpass filter was selected so as to cover the whole bandwidth of the transducer and not to lose important information of high range of frequency domain, but also to eliminate noise. 

Frequency spectra of the signal measured was calculated according to:(10)U(f)=FT[u(t)],
where FT is a denoted Fourier transform.

Frequency spectrum was filtered using band-pass filter.
(11)$$U_{F}\left(f\right)=U\left(f\right)\cdot H\left(f\right),$$
where *H(f)* is a filter transfer function. 

The filter characteristic was calculated according to:(12)$$H\left(f\right)=\{\begin{array}{ll} 1, & f_{1}+\frac{\Delta f}{2}<f<f_{1}+\frac{\Delta f}{2}\\ \sin\left(\frac{\left(f-f_{1}\right)\cdot \pi }{2\cdot \Delta f}\right)+0.5, & f_{1}-\frac{\Delta f}{2}<f<f_{1}+\frac{\Delta f}{2}\\ \sin\left(\frac{\left(f_{2}-f\right)\cdot \pi }{2\cdot \Delta f}\right)+0.5, & f_{2}-\frac{\Delta f}{2}<f<f_{2}+\frac{\Delta f}{2}\\ 0, & \mathrm{in}\;\mathrm{other}\;\mathrm{cases} \end{array},$$
where ·f is the width of the fronts of filter function, $f_{1}$
*=* 2.2 MHz is the lower cut-off frequency, $f_{2}\;$
*=* 14.3 MHz is the upper cut-off frequency, and f is the frequency.

The reconstruction of the filtered signal was performed using inverse Fourier transform: (13)uF(t)=Re(FT−1[UF(f)]),
where Re denotes the real part, and ${\mathrm{FT}}^{-1}$ denotes inverse Fourier transform. 

After the filtering, the alignment of the signals is performed according to the arrival time of surface reflection in the following steps:
The arrival time of surface reflection at a set threshold is determined according to the equation:(14)$$t_{n_{1},k}=\min\left\{\arg\left[u_{k}\left(t_{n_{1}}\right)>U_{th}\right]\right\},$$
where $u_{k}\left(t_{n_{1}}\right)$ is a digitized signal, k = from 1 to *K*, *K* is a number of signals, and $t_{n_{1}}$ is the time of the first sample which is exceeding the threshold $U_{th}$.The first transition through the zero crossing point in the signal was found according to equation:(15)$$t_{0,k}=\min\left\{\arg\left[u_{k}\left(t_{n}\right)=0\right]\right\},$$
where $t_{n}$>$t_{n_{1}}$, $u_{k}\left(t_{n}\right)$ is a digitized signal of all time moments $t_{n}$ exceeding $t_{n_{1}}.$All signals are shifted according to equation:(16)$$u_{k}^{\prime }\left(t_{n}\right)=u_{k}\left(t_{n}+t_{0,k}\right).$$

A detailed description of the zero-crossing technique used was presented in the work of phase velocity measurement of Lamb waves [43]. 

As a result, a new B-scan with all signals aligned was obtained. 

Furthermore, the time gate is set at surface reflection. Two signals of perfect bonding and debonding between two adhesive tapes were modeled and compared to each other. The comparison is required in order to determine the time intervals of the signal with the high change in amplitudes.

Peak-to-peak amplitudes in selected time intervals are calculated according to:(17)$$M_{n}=\max\left(u\left(t\right)\right)-\min\left(u\left(t\right)\right),$$
where *t* belongs to the selected time intervals, and *n* = from 1 to *N* is the number of multiple reflections from the boundary.

The adhesion quality can be identified from the decay law of multiple reflections of the signal. Therefore, coefficients representing the ratio of peak-to-peak amplitudes in the selected time intervals were calculated in order to study and compare how fast the signal decays: (18)$$K_{1}=\frac{M_{n}}{M_{n+1}},$$
(19)$$K_{2}=\frac{M_{n+1}}{M_{n}},$$
where $M_{n}$ represents the peak-to-peak amplitudes in the selected time intervals, and *n* = from 1 to *N* is the number of multiple reflections from the boundary.

### 2.4. Experimental Set-Up of Ultrasonic Testing

The immersion pulse echo technique of ultrasonic non-destructive testing was used for an experimental investigation of a single-lap joint of adhesively bonded dissimilar materials. The inspection was performed in an automated immersion ultrasonic testing system TecScan (TecScan Systems Inc.) with TecView software for 3D testing. A 10 MHz immersion focused transducer Olympus V375-SU with 2 inches (50.8 mm) focus distance manufactured by Olympus Scientific Solutions Americas Inc. was used for experimental inspection as a receiver/transmitter. The transducer and the component were immersed into the tank of water. The sample was placed on the turntable and adjusted. Using system scanners, the transducer was positioned perpendicularly to the surface of the object and focused on the interface between metal and composite materials in order to increase detectability in this area. Scanning of the component was performed along *y* and *z* axes with the step of 0.1 mm after the scan area was set in the software. The distance between the transducer and component surface was 43.5 mm. The inspection set-up is shown in Figure 10. 

Aluminum side inspection was selected, since CFRP is a more attenuating material due to its material characteristic and laminate structure [44]. However, the thickness of the aluminum layer is quite thin, which can lead to the overlapping of different signal reflections [8,35,36]. 

## 3. Results and Discussions

In this section, results of the experimental investigation of the component, disbond detection complexity, and main influencing parameters are presented and discussed. The method in order to increase detectability was applied, and received results are presented. 

### 3.1. Comparison of Modeled and Experimental Signals 

A layered structures inspection technique based on the model analysis of multiple reflections was performed. The parameters of the model were optimized in order to obtain the closest fit of modeled signals to the experimental ones and are presented in Table 4.

A comparison of the modeled and experimental signals reflected from the adhesively bonded dissimilar materials without any defect is shown in Figure 11. The time of signal reflections from the sample boundaries was calculated according to Equation (7). 

The modeled signal matches the experimental one along the time of reflections from the component boundaries. Reverberation influence is observed in multiple interface reflections in the time interval between two neighboring reflections of the aluminum/adhesive boundary in both modeled and experimentally received signals. The time instances corresponding to the arrival time of the signals reflected by each boundary were calculated and indicated in A-scan. It can be seen that reflections from different boundaries in the interface line are overlapping at time instances of reflections from aluminum/adhesive, adhesive/GFRP, and GFRP/CFRP boundaries. 

An A-scan comparison of two modeled signals of perfect bonding and debonding between two adhesive tapes is shown in Figure 12. 

According to the modeling, it can be seen from the A-scan that a significant change in amplitude is observed after the second time instance t_1ad,GFRP_ of the signal at the interface reflection time interval: t_1i_:t_2i_. Therefore, part of the signal after a particular time instance in the t_1i_:t_2i_ time interval was selected from the experimental data to create a C-scan image of the sample. The same was performed for multiple reflections from the interfaces t_2i_:t_3i_, t_3i_:t_4i_, and t_4i_:t_5i_. As a result, selected time intervals that will be used in the processing part of the experimental data are presented in Table 5.

### 3.2. Experimental Investigation Results and Demonstration of the Improvement of Disbond Detectability

After experimental investigation of the layered sample using the selected technique, the C-scans of the top view of the sample (length and width with *x* and *y* axes) were created. The C-scans of the interface reflection of time interval t_1i_:t_2i_ are shown in Figure 13. 

Carrying out the experimental inspection, the influence of the surface curvature can be observed. The highest time difference influenced by curvature is about 1 microsecond. From the C-scan of the t_1i_:t_2i_ interface reflection (Figure 13) presented, it was observed that the components curvature has a high impact on disbond detectability. As a result, only the biggest defect can be barely distinguished, but two smaller ones were not detected. The biggest disbond has a higher amplitude of the signal compared to the signal of the perfectly bonded area. In the case of two smaller defects, there is no change in signals amplitude. 

The data of experimental investigation were processed using the 1st and 2nd step of the proposed algorithm for disbond detectability improvement. The resulting C-scans and A-scan with a selected time intervals of multiple interface reflections are shown in Figure 14. The influence of the component curvature on disbond detectability was partially eliminated. All three disbonds were detected analyzing multiple reflections from the aluminum/adhesive boundary of the component of indicated time intervals. In the case of the biggest defect, the detectability is increasing by the analysis of repeated multiple reflections from the component boundary. The fourth interface reflection of the t_4i_:t_5i_ time interval (Figure 14e) presents the best results of all defects detection. However, two smaller defects can be barely distinguished. Additionally, the perfectly bonded area has a trend of increased amplitude that interferes with the detection of the disbonds.

Furthermore, C-scans were created by applying the 4th and 5th steps of the developed processing algorithm to evaluate time intervals with the significant change in amplitude (M_1_–M_4_). The A-scan with indicated time intervals as well as the created C-scans are shown in Figure 15. As a result, all defects are identified. Smaller disbonds are highlighted more clearly. Moreover, the detectability of all three disbonds increases with the analysis of multiple reflections. 

C-scans created by applying the technique of calculated ratio coefficients, which corresponds to step 6 of the developed algorithm, are shown in Figure 16. From the presented C-scans, it can be observed that the detectability of defects was improved. All three disbonds are located and identified clearly. The shapes of three defects are more distinct. 

## 4. Conclusions

The investigation carried out has demonstrated the most critical factors influencing the detectability of disbonds in dissimilar material joints such as defect depth location in the adhesive layer, incidence angle (component curvature), and thickness of the aluminum layer. 

It was shown that in order to increase detectability, the time alignment of the signals according to reflection from the top surface is required, even in the case of a slight curvature of the sample. Afterwards, the detailed analysis of multiple reflections from the adhesive bonding interface of the component should be performed. 

Exploiting results obtained during the investigation, a method for the improvement of detectability was proposed. This method is based on a developed advanced signal processing algorithm that includes filtering, time alignment, determination of intervals with the largest amplitude changes, calculation of peak-to-peak amplitudes, and estimation of ratios in selected time intervals. The proposed method enabled detecting all three disbonds in the joint of dissimilar materials with improved detectability.

## Figures and Tables

**Figure 1 sensors-21-03048-f001:**
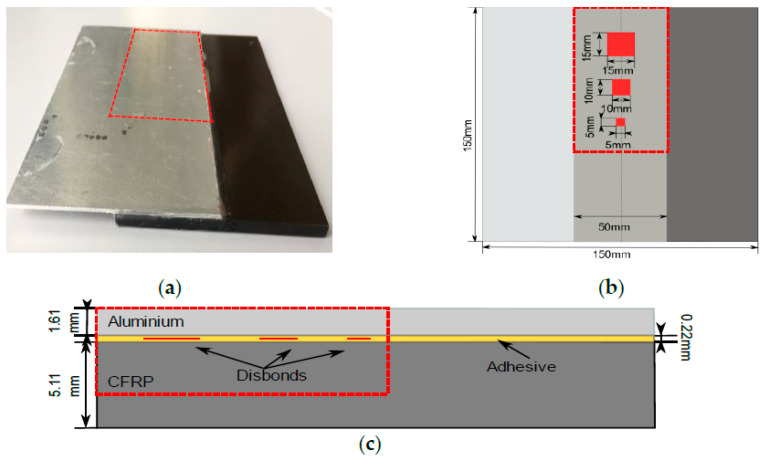
Single-lap joint of dissimilar materials under investigation: (**a**) Photo with indicated region of disbonds location; (**b**) Top view scheme of the component and defects dimension; (**c**) Cross-sectional view of the component and defects.

**Figure 2 sensors-21-03048-f002:**
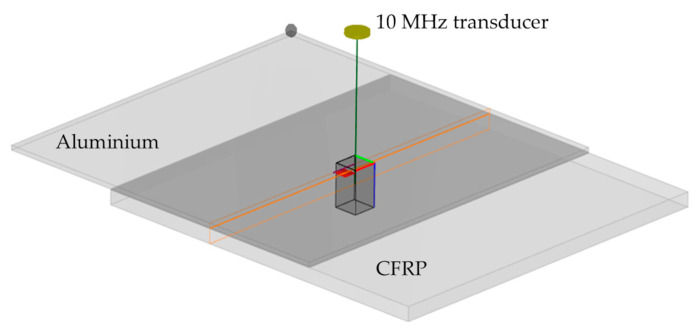
Ultrasonic pulse echo inspection simulation set-up of the sample of single-lap joint.

**Figure 3 sensors-21-03048-f003:**
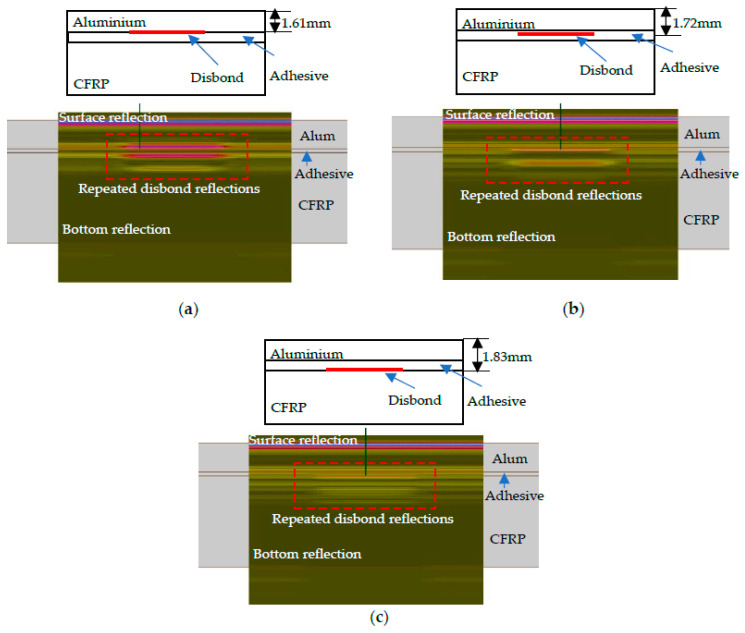
Disbond location and B-scans of the inspection simulation: (**a**) Disbond location between aluminum/adhesive; (**b**) Disbond location in the middle of the adhesive layer; (**c**) Disbond location between adhesive/composite.

**Figure 4 sensors-21-03048-f004:**
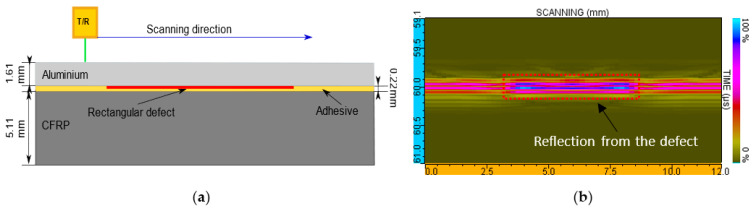
Calibration of inspection simulation on a rectangular-type defect to determine the reference amplitude required for metamodel estimation: (**a**) Set-up of inspection simulation; (**b**) Resulting B-scan.

**Figure 5 sensors-21-03048-f005:**
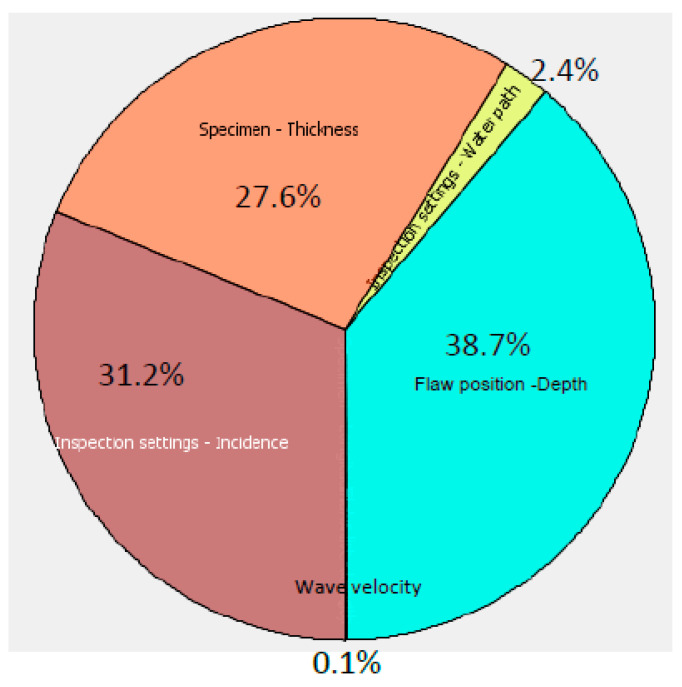
Sensitivity analysis of inspection and defect parameters.

**Figure 6 sensors-21-03048-f006:**
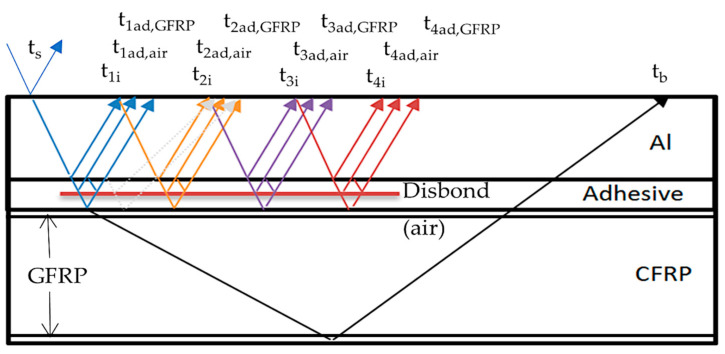
Path of ultrasonic wave propagation and reflection from the boundaries.

**Figure 7 sensors-21-03048-f007:**
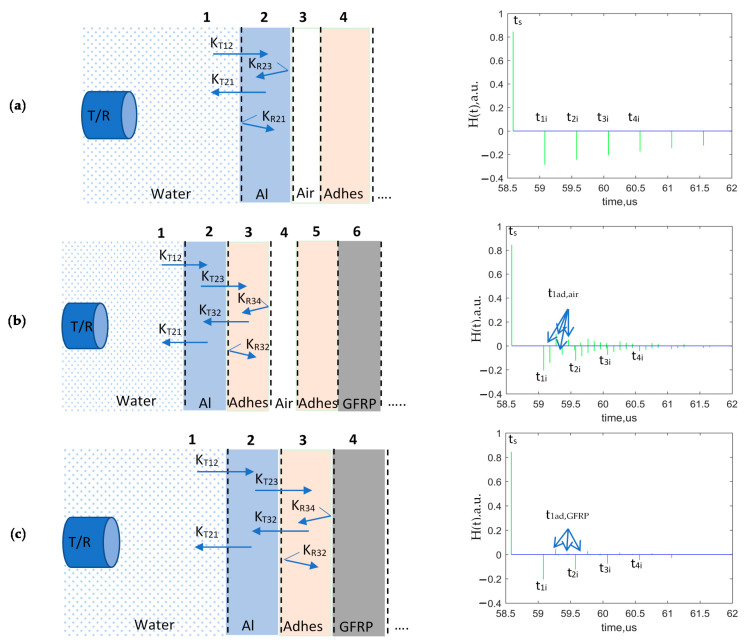
Three different cases of analyzed boundary conditions: (**a**) Total debonding between aluminum and adhesive layer; (**b**) Total debonding in adhesive layer; (**c**) Perfect bonding. The layers of the sample are presented not in scale.

**Figure 8 sensors-21-03048-f008:**
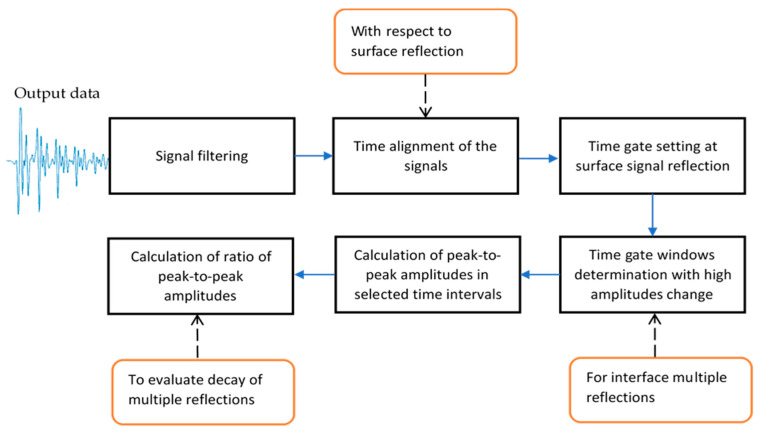
Functional block diagram of main steps of the developed algorithm.

**Figure 9 sensors-21-03048-f009:**
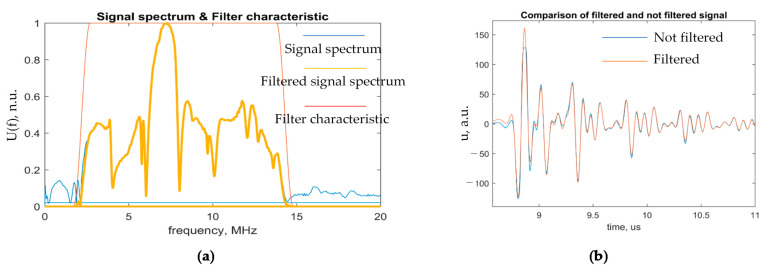
Signal filtering: (**a**) Signal spectrum and filter characteristics; (**b**) Comparison of filtered and not filtered signals.

**Figure 10 sensors-21-03048-f010:**
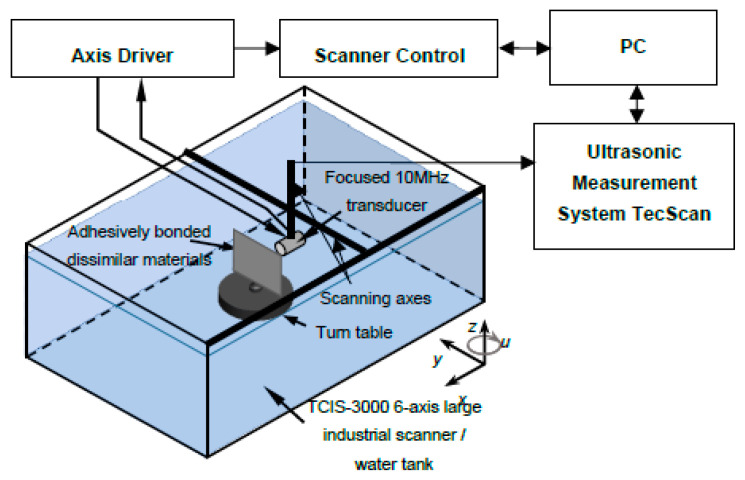
Ultrasonic pulse echo inspection set-up of a single-lap joint of adhesively bonded dissimilar materials.

**Figure 11 sensors-21-03048-f011:**
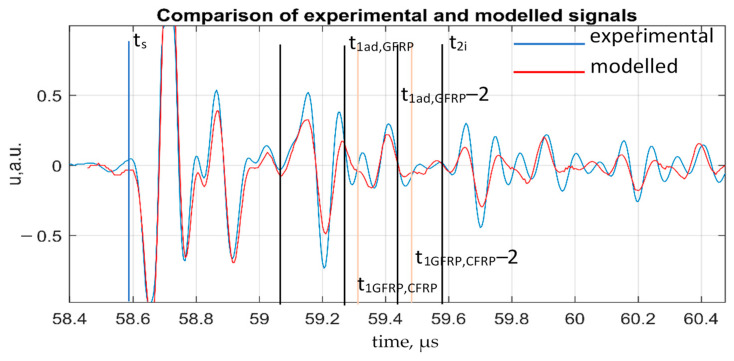
Comparison of modeled and experimental signals received after wave propagation through the component of perfect bonding (t_s_—time of reflection from the component surface, t_1i_—time of reflection from the interface of aluminum and adhesive, t_1ad,GFRP_—time of reflection from the interface of adhesive and GFRP, t_GFRP,CFRP_—time of reflection from GFRP/CFRP, t_2i_, t_2ad,GFRP_, t_2GFRP,CFRP_—corresponding repeated reflections).

**Figure 12 sensors-21-03048-f012:**
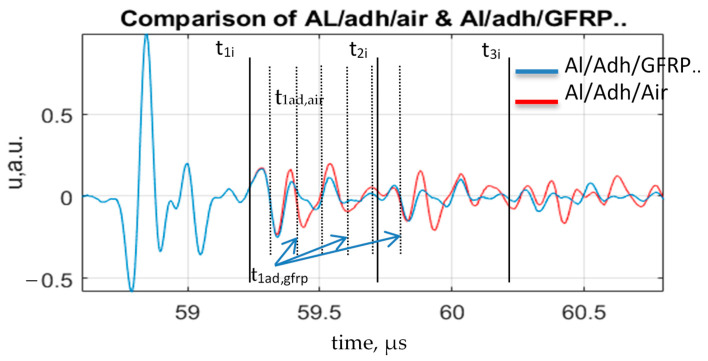
Comparison of two modeled signals: perfect bonding (Al/Adh/GFRP/CFRP/GFRP) and total debonding (Al/Adh/Air).

**Figure 13 sensors-21-03048-f013:**
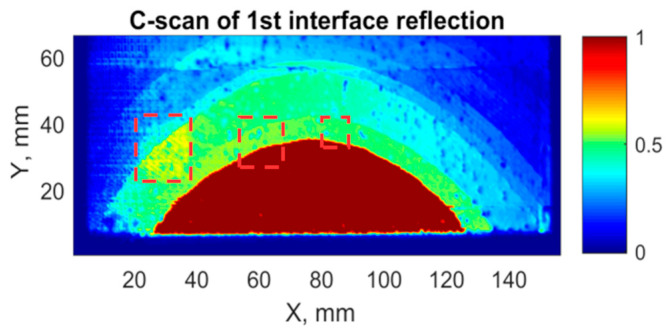
C-scan of 1st interface reflection, time interval (t_1i_:t_2i_).

**Figure 14 sensors-21-03048-f014:**
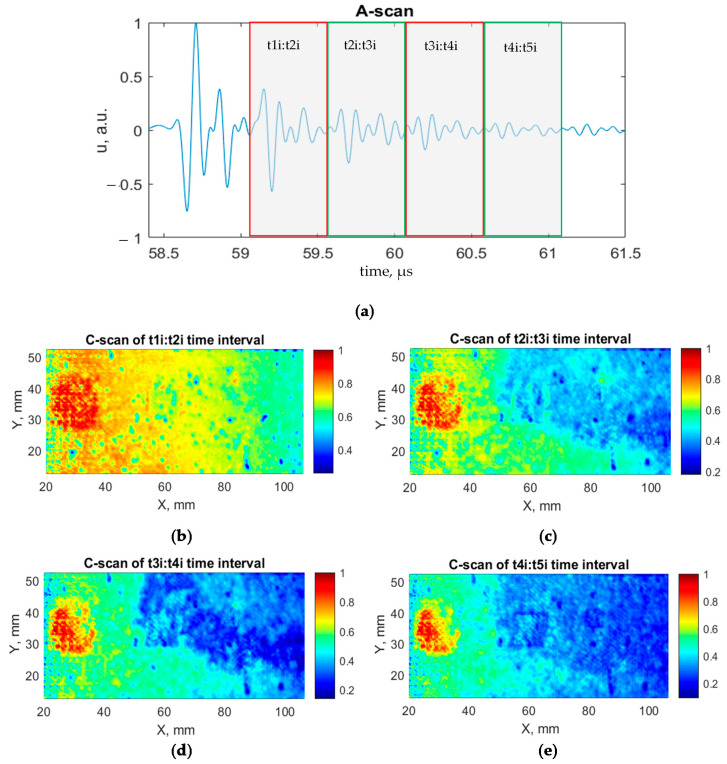
A-scan and C-scan of repeated interface reflections: (**a**) A-scan with indicated time intervals selected to display C-scans; (**b**) C-scan of t_1i_:t_2i_ time interval; (**c**) C-scan of t_2i_:t_3i_ time interval; (**d**) C-scan of t_3i_:t_4i_ time interval; (**e**) C-scan of t_4i_:t_5i_ time interval.

**Figure 15 sensors-21-03048-f015:**
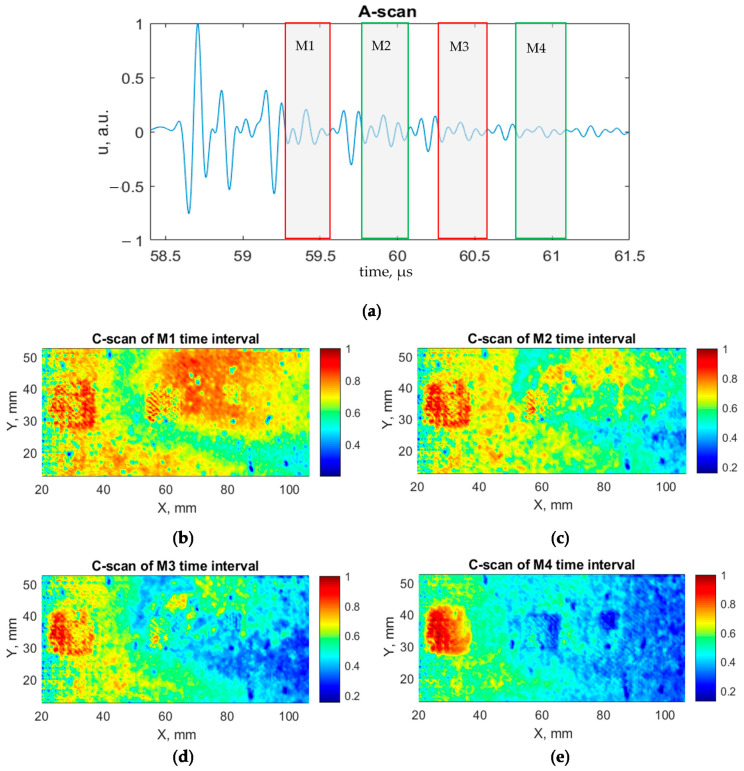
A-scan and C-scans: (**a**) A-scan with indicated time intervals selected to display C-scans; (**b**) C-scan of M_1_ time interval; (**c**) C-scan of M_2_ time interval; (**d**) C-scan of M_3_ time interval; (**e**) C-scan of M_4_ time interval.

**Figure 16 sensors-21-03048-f016:**
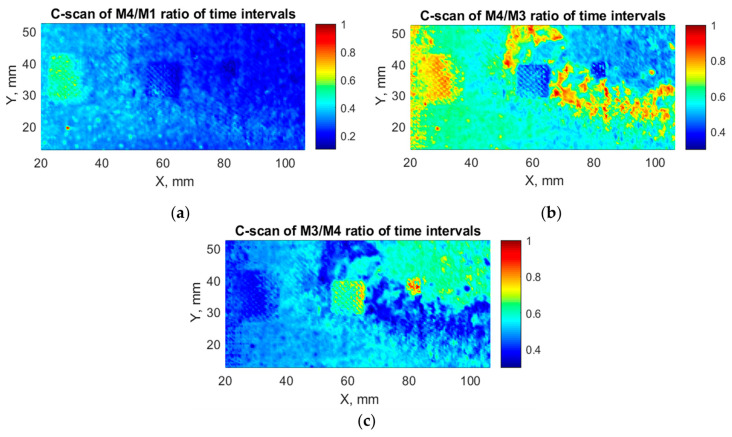
C-scans of calculated ratio: (**a**) C-scan of M_4_/M_1_ ratio; (**b**) C-scan of M_4_/M_3_ ratio; (**c**) C-scan of M_3_/M_4_ ratio.

**Table 1 sensors-21-03048-t001:** Variation range of influential parameters.

Parameters	Mean Value	Variation Range	Statistical Distribution Law
Aluminum longitudinal wave velocity	6363 m/s	[6313 m/s; 6500 m/s]	Normal
Defects dimension along bondline *x* axis	2.625 mm	[0.25 mm; 5 mm]	Constant/Characteristic value
Defects depth in adhesive layer	1.72 mm	[1.61 mm; 1.83 mm]	Uniform
Water path between transducer and object surface	43.5 mm	[43.5 mm; 44.03 mm]	Normal
Thickness of aluminum layer	1.61 mm	[1.60 mm; 1.62 mm]	Normal
Incidence angle	0°	[−3°; +3°]	Normal

**Table 2 sensors-21-03048-t002:** Values of sensitivity and proportion of parameters impact.

Parameters	Sensitivity	Proportion Value, %
Wave velocity	0.16	0.11
Defects depth	55.64	38.7
Water path	3.42	2.4
Thickness of aluminum layer	39.65	27.6
Incidence angle	44.77	31.2

**Table 3 sensors-21-03048-t003:** Amplitudes of multiple reflections in each particular layer.

Amplitude of Multiple Reflections	Aluminum/Air	Adhesive/Air	Adhesive/GFRP
A_1_	−0.2915	−0.1451	0.0468
A_2_	−0.2424	0.1020	0.0106
A_3_	−0.2016	−0.0717	0.0024
A_4_	−0.1676	−0.0504	0.00054

**Table 4 sensors-21-03048-t004:** Defined material and medium characteristics to model the signal closest to experimental results.

Material	Thickness, mm	Density, kg/m^3^	Ultrasound Velocity, m/s
Water	43.5	997.98	1485
Aluminum	1.61	2710	6500
Adhesive	0.22	1270	2315
Glass prepreg	0.06	1900	3000
CFRP	4.99	1800	2800
Air	10	1225	330

**Table 5 sensors-21-03048-t005:** Selected time intervals with significant change in amplitude.

Peak-to-Peak Amplitude in the Selected Time Intervals	Selected Time Intervals
M_1_	t_1ad,GFRP_: t_2i_
M_2_	t_2ad,GFRP_: t_3i_
M_3_	t_3ad,GFRP_: t_4i_
M_4_	t_4ad,GFRP_: t_5i_

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
