# Peer review of "Novel Processing Algorithm to Improve Detectability of Disbonds in Adhesive Dissimilar Material Jointsâ€"

_sensors, 2021, doi:10.3390/s21093048_

Round 1
Reviewer 1 Report
The paper proposes a methodology to improve the detectability of disbond in adhesive joint of Aluminum and CFRP. The paper content, and its scientific relevance sound good, but, unfortunately, the paper is not well organized. As consequence, the significance of the proposal is tricky to be highlighted. In the following, some comments and suggestions.
- The paper must be deeply revised also reorganizing the proposed sections and reducing the paper length (deleting not significant parts) with the aim to improve both readability and clearness.
- In the reorganization of the paper section try to better schematize the paper content.
- Up to page 10 it is not so clear if you are presenting the results of simulations or you are just defining all the quantities needed to make the simulation/experimental tests presented in the successive parts of the paper. If you are presenting the results of simulations, where are the results? There is a lack of analysis and discussion of the simulated results.
- Simulations are usually done to obtain results that is not possible (or it is difficult) to obtain experimentally. In some other cases are used to design a system or to make a deep sensitivity analysis. Typically, the simulation results are more exhaustive and complete of the experimental results that are used to confirm the goodness of the results obtained in simulations. You must better highlight the aim of your simulations showing the results and detailing discuss them.
- The sensitivity analysis reported in figure 5 is not so clear. The variability range of the considered parameters (together with the choice of them) have to be reported in the paper. The final results reported in figure 5 must be accompanied by data and results (table, graph) analyzed and discussed.
- Page 7 line 247. I suggest to define on this line kr and kt. Something like: “…as well as the reflection (kr) and transmission (kt) coefficient were calculated.”
- From page 7 to the end of the paper there are a lot of errors in the recall to equations and figures. As for examples:
- Page 7 line 257 – equation should be 6 and 7 (not 5 and 6)
- Page 8 line 270 – Figure 6 not 5
- Page 8 line 280 – Figure 7 not 6
- Page 9 line 295 – Figure 8 not 7
- … and so on
- Figure 7 should be better commented and descripted. Please also define who is “H(t) a.u.”
- Page 10 lines 298-300 – Always to stress the importance to clarify the meaning and the scope of the simulations and the analysis made (see previous comments), you say here that an optimization was performed by selecting the value of parameters …. Which ones? In which range? How you made the optimization? What are the final value after the optimization?
- Figure 9 – Describe in detail the experimental setup.
- Page 11 lines 339-345. I suggest to create a functional block diagram of the proposed algorithm
Reviewer 2 Report
The manuscript, "Improvement of Disbond Detectability in Adhesive Dissimilar Material Joints" by Damira Smagulova et al., describes detecting adhesive voids/debonding using improved ultrasonic signal processing algrithms. The manuscript provides solid simulation and experimental results. The manuscrip should get published eventually after addressing my comments:
- Secton 2.6 (line 315-323), more detailed experimental information is required. What's the model and manufacture of the key components in the experiment, like ultrasonic ultrasonic transducer, receiver, rotation stage? What special protection techniques are required to put all the instrument under water?
- Fig. 5, a more detailed description about Metamodel is required. For example, how the pie chart is plotted and how to define the percentage of each category?
- Fig. 7, in the time vs H (t) plot, what's the spikes in the time? Why some of them are positive and some are negative? Legend is required to show the meaning of each color in the plot.
- Fig. 10, how the bandwidth of the bandpass filter is defined?
- Fig. 11, is it possible to overlap the cross-section vs B-scan results? This will help the reader to better understand the results. Meanwhile, what's the defect depth used in the simulation?
- Fig. 15-17, what's the meaning of Y,mm in the y-axis? By visual inspection the detection results, the square shape of defects are more obious after algrithm modification. Are there any other parameters (like skewness, edge sharpness, etc) that can be used to compared the detection improvement other than visual inspection?
- From Fig. 17, it looks the biggest square didn't get detected completely compared with the middle/small square defects in terms of square shape. Any explanation to this result?
- What's the resolution of this method to detect the defects? In terms of defect size, depth, position, etc.
- In all the plots, it is not proper to use "us" to describe "micro second". The correct geek letter should be used.
Reviewer 3 Report
the current study investigates methods of improving the detectability of defects which are present in adhesively bonded dissimilar material using immersion pulse-echo ultrasonic technique. For that the authors carry out simulations to understand the ultrasonic wave propagation with different defect location in bondline in layers of disks made from different materials.
The title does not reflect at all what has been in this study and therefore must be updated (include the words improving the detectability of defects in dissimilar materials …..etc
The abstract needs to be improved, it does not read well, there is no mention of what has been done exactly and what were the main findings from this study. Please consider reviewing the abstract and highlight the novelty, major findings and conclusions.
The authors are encouraged to add a list of nomenclature at the start or end of the manuscript
Line 215-222 write it in a paragraph and not in bullet points as this way it reads more like a manual, check this issue elsewhere
Line 403-404 “due to attenuation in adhesive layer and overlapping of signal reflections from adhesive top, bottom and disbond” can you support this claim with a reference, also what about past studies did they report similar findings of different from yours? If not then explain and support with references
Line 412 can you support this claim with a reference
Line 424 why? Please explain and support with references
Line 492-502 combine in one larger paragraph, please check this issue everywhere else and avoid writing smaller paragraphs
The results are merely described and is limited to comparing the experimental observation. The authors are encouraged to include a discussion section and critically discuss the observations from this investigation with existing literature
Round 2
Reviewer 1 Report
The authors have tried their best to improve the paper quality. Even if it can be still improved for clearness and readability, it is now ready to be published.
Author Response
Thank you for your valuable comments; we tried to consider them as much as possible during first review. We have performed additional editing to improve the quality of the paper during second review.
Reviewer 3 Report
- As in previous revision, The authors should add a list of nomenclature at the start or end of the manuscript which includes all symbols and letters used in the manuscript
- Paper needs English editing and spelling check.
